# In Situ Formation of MoS_2_ on the Surface of CF to Improve the Tribological Properties of PUE

**DOI:** 10.3390/ma16175773

**Published:** 2023-08-23

**Authors:** Cunao Feng, Yu Guo, Xiaowei Li, Yang Cao, Qiuxue Kuang, Minghui Zhang, Dekun Zhang

**Affiliations:** School of Material Science and Physics, China University of Mining and Technology, Xuzhou 221116, Chinaxwli@cumt.edu.cn (X.L.); dkzhang@cumt.edu.cn (D.Z.)

**Keywords:** PUE, CF, MoS_2_, mechanical properties, friction performance

## Abstract

The roller is an important part of the belt conveyor used in coal transportation. Due to the harsh environment of coal mines, the rollers are in a state of high load and high friction for a long time, which causes wear failure and has a serious impact on the reliability and safety of the equipment. In order to prepare roller material with excellent bearing performance and friction performance, CF/PUE composites were prepared by pouring method with polyurethane as the matrix and carbon fiber as reinforcement. Due to the low surface activity of unmodified carbon fibers and poor bonding performance with the matrix, MoS_2_ was generated on the surface of carbon fiber by the in situ generation method in this paper. It was found that the mechanical properties of MoS_2_/CF/PUE composites were better when the CF content was 0.3 wt%. The Shore hardness reached 92.2 HA, which is 10% higher than pure polyurethane. The tensile strength was 38.44 MPa, which is 53% higher than pure polyurethane. The elongation at break was 850%, which is 16% higher than pure polyurethane. The maximum compressive stress was 2.32 MPa, which is 42% higher than pure polyurethane. The friction coefficient was much lower than that of pure PUE composites, the friction coefficient was 0.284, which is 59% lower than pure polyurethane.

## 1. Introduction

The roller is the most critical component of the belt conveyor, which is located at the bottom of the upper and lower conveyor belts and plays a role in supporting the conveyor belts and reducing friction resistance [1]. The number of rollers under each conveyor belt is relatively large, and the longer the conveyor belt is, the greater the demand for rollers. Its cost accounts for about 35% of the total cost of the belt conveyor, and it can bear more than 80% of the load, and the energy consumption accounts for 70% of the total energy consumption [2,3]. At present, most of the rollers used in coal transportation are made of metal materials. This kind of roller has a large mass and will consume a lot of energy in the process of coal mining. Moreover, the roller is exposed to a humid environment for a long time or is surrounded by a large amount of coal dust and corrosive gas, it becomes easily rusted, locked, and severely worn, which adversely affects the reliability and safety of the equipment, and also causes significant economic losses and energy waste. At present, there are few rollers with high quality and long service life made in China, the design standards for rollers are also too loose, and they are mostly positioned as cheap consumables. The normal operation of equipment is often realized by increasing the weight of rollers or replacing rollers many times, which not only consumes a lot of manpower and material resources, but also loses a lot of effective production time, resulting in great losses [4].

Polyurethane (PUE) is a kind of block copolymer, which is a lightweight polymer material with corrosion resistance, high wear resistance, a low friction coefficient, and low cost [5,6,7]. At present, it is widely used in coal, metallurgy, construction, and other industries, and is an excellent roller material. However, polyurethane has poor toughness, poor impact resistance, and is prone to aging, making it difficult for polyurethane to meet corresponding requirements in some unique applications [8,9]. Carbon fiber (CF) is a kind of fiber with a carbon content greater than 90%, which has the advantages of being lightweight, having high strength, a large modulus, good thermal performance, friction resistance, and fatigue resistance, so it is widely used in aircraft, automobiles, the military, chemicals, and other fields [10,11,12,13,14,15]. The excellent friction performance of CF itself causes the CF-reinforced PUE composite to have a low friction coefficient, [16] and the graphite scrap during the friction process can realize the self-lubrication of the material. Therefore, chopped CF or continuous CF-reinforced PUE is commonly used at home and abroad [17]. However, CF has a smooth surface and strong polar groups. When combined with PUE, it is difficult to form a solid interface between the two, which affects the mechanical and frictional properties of the composite material. In addition, CF is prone to breakage when bent. In order to increase the bonding force with the PUE matrix, the surface of CF needs to be modified to increase the surface roughness and surface activity of CF and improve the mechanical and frictional properties of composite materials [18,19]. MoS_2_ is generally extracted by the mineral chalcocite, which is not affected by dilute acid and oxygen, and its performance is relatively stable [20]. Molybdenum atoms and sulfur atoms form a planar layer by covalent bond. Different layers are connected by weak interaction. This structure causes MoS_2_ to have a very low friction coefficient, about 0.04, which is often used for bearing lubrication and material modification, but MoS_2_ has poor thermal conductivity [21,22]. In research on MoS_2_, researchers have paid attention not only to the quality of MoS_2_, but also to the dispersion of MoS_2_ in the polymer matrix. The dispersion of MoS_2_ in the polymer matrix plays a key role in the properties of MoS_2_/polymer composites [23]. Therefore, it is necessary to add fillers with high strength, fatigue resistance, and good thermal conductivity to the polyurethane matrix to improve the performance of polyurethane composite materials.

In this paper, MoS_2_ was generated in situ on the surface of carbon fiber by in situ generation method to prepare MoS_2_/CF composites (Figure 1). By adding modified CF to improve the properties of PUE, we studied the effects of different contents of modified CF on the mechanical and tribological properties.

## 2. Experimental Materials and Methods

### 2.1. Experimental Materials and Equipment

The equipment materials used in the experiment are shown in Table 1.

### 2.2. Preparation of PUE and Its Composites

#### 2.2.1. Preparation of PUE

As shown in Figure 2. Casting PUE was prepared by using an MDI prepolymer and 1,4-butanediol chain extender as raw materials. Firstly, the solid MDI prepolymer was heated in the oven at 85 °C, and the mold was heated in the oven at 120 °C after spraying the release agent. After the prepolymer melts, weigh an appropriate amount and pour it into the beaker, vacuum the vacuum tank until no obvious bubbles are generated, and then put it into the oven for heating for 30 min. According to the mass ratio of prepolymer: chain extender = 10:1, weigh the chain extender 1,4-butanediol, pour it into a beaker with prepolymer, rapidly stir for 1 min, and then vacuumize for 20 s. Quickly pour the liquid PUE into the mold, put it into the oven at 120 °C, and cover the mold after gelling. After curing in an oven at 120 °C for 1 h, take out the sample and put it into an oven at 85 °C for 22 h. The test was carried out after 7 days at room temperature.

#### 2.2.2. Preparation of MoS_2_/CF

After drying a certain amount (0.1, 0.3, 0.5, 0.7, 0.9%) of CF, put it into a nitric acid solution for ultrasonic treatment for 60 min. After the treatment, rinse the reacted CF with deionized water many times, and dry it at 100 °C for standby. Subsequently, a certain amount of thiourea (CSN_2_H_4_) and ammonium molybdate (NH_4_)_6_Mo_7_O_24_•4H_2_O) are then added to 100 mL of deionized water, and dispersed by ultrasound for 30 min to accelerate dissolution. Then, add an appropriate amount of CF treated with nitric acid into the mixed solution, and disperse it by ultrasonic wave for 30 min to obtain a uniformly mixed CF mixture. Finally, the mixed solution is poured into a reaction kettle with a capacity of 200 mL, and after sealing, it is placed in an oven at 230 °C for 24 h. After centrifugation, cleaning and drying at 100 °C for 24 h, carbon fibers with molybdenum disulfide on the surface are obtained [24].

#### 2.2.3. Preparation of MoS_2_/CF/PUE Composites

We chose and used an MDI prepolymer, 1,4-butanediol chain extender, and modified CF as raw materials to prepare MoS_2_/CF/PUE composites. Firstly, heat the solid MDI prepolymer at 85 °C in the oven and dry the modified CF powder in the oven. Heat the mold at 120 °C in the oven after spraying it with a release agent. Weigh a certain amount of modified CF and melted prepolymer into a beaker, and stir until they are evenly mixed. Vacuumize the vacuum tank until there is no obvious bubble, and put the mixture into the oven again for 30 min. According to the mass ratio of prepolymer: chain extender = 10:1, weigh the chain extender 1,4-butanediol, pour it into a beaker with prepolymer, rapidly stir for 1 min, and then vacuumize for 20 s. Quickly pour the liquid-modified CF/PUE composite into the mold, put it into the oven at 120 °C, and put it into the mold after gelling. After curing in an oven at 120 °C for 1 h, the samples are taken out and placed in an oven at 85 °C for 22 h. Testing is performed after 7 days at room temperature.

### 2.3. Hardness Test

PUE elastomer is a rubber material, and the Shore hardness A is tested according to the national standard GB/T 531.2-2009. The hardness test of the prepared composite material is carried out, the three points with a large distance are taken for measurement, and the average value is obtained to obtain the hardness value.

### 2.4. Tensile Property Test

The PUE is cast into a dumbbell-shaped standard tensile specimen, as shown in Figure 3. Use the universal tensile testing machine to carry out the tensile test according to GB/T528-2009, 5 samples in each group, take the average value. The stretching rate is 200 mm/min.

### 2.5. Compression Performance Test

PUE is a kind of material with good elasticity. During the compression test, the PUE elastomer will undergo large elastic deformation but the structure is hard to damage. A WDW-5 microcomputer-controlled electronic universal testing machine was used in the experiment. According to the measurement standard of vulcanized rubber or thermoplastic rubber (ISO 7743:2007), the sample was compressed and deformed by 10%, and the compression stress under this deformation was taken as the compression strength of PUE. The sample and its dimensions are shown in Figure 4, where a = 11.5 mm and h = 6.3 mm. The compression rate is 2 mm/min, each sample is tested five times, and the average value is taken as the compression performance index of the material.

### 2.6. Tribological Performance Test

In this experiment, a UMT-3 friction and wear testing machine was used to study the friction and wear of PUE composite materials; we used sliding friction. The specific parameters of the experiment were set, as the fixed load was 100 N, the sliding speed was 5 mm/s, the reciprocating frequency was 1 Hz, and the experiment time was 120 min. The upper sample was a steel ball with a diameter of 6 mm, and the lower sample was a cuboid PUE composite material sample of 48 × 22 × 5 mm. The matching pair of the upper and lower samples is shown in Figure 5.

### 2.7. Wear Morphology Analysis

Before the friction and wear test, the surface of the sample was smoothed, washed with anhydrous ethanol, and then placed in a 60 °C blast-drying oven to dry. The macroscopic wear morphology of the samples was observed and analyzed using a VW-9000 high-speed camera system. Select the wear location with obvious features, use the coarse focus screw knob and fine focus screw knob to respectively focus by adjusting the magnification, adjust the aperture to the appropriate brightness after the object is displayed on the computer side, then take pictures, and name and save in the appropriate location. Take multiple photos of multiple feature positions, and repeat the above operations for each proportioning sample in turn.

### 2.8. X-ray Diffraction Analysis (XRD)

An X-ray diffractometer (XRD) can accurately identify the phase composition in the sample. In order to prove whether MoS_2_ is formed in CF, XRD was used to analyze the phase of CF before and after MoS_2_ modification. We used a D8ADVANCE X-ray diffractometer from Bruker, Germany, to test the CF and CF/MoS_2_ composites. Test conditions: copper target Kα radiation, filament voltage of 50 kV, current of 30 mA, continuous scanning and scanning speed set to 0.2 sec/step, and scanning range set to 5°~80°.

### 2.9. Scanning Electron Microscope Analysis (SEM)

Scanning electron microscopy (SEM) was used to observe the morphology of CF and the tensile and frictional morphologies of PUE composites. The tensile fracture sample of the PUE composite material was cut at about 1 mm of the tensile fracture surface, and the microscopic morphology of the tensile fracture surface was observed and the fracture of the tensile sample was analyzed. The fracture mechanism and friction and wear mechanisms of the CF and PUE composites before and after modification were analyzed. In this experiment, the G-301 scanning electron microscope produced by Hitachi, Japan, was used for testing.

## 3. Results and Discussion

### 3.1. Microstructure Characterization of PUE/CF/MoS_2_ Composites

Figure 6 is the X-ray diffraction pattern of CF and MoS_2_-modified CF. The XRD pattern can show the diffraction peak position and the intensity of crystal or semi-crystal materials, from which information such as crystal plane spacing and lattice parameters can be deduced. It can be seen from the curve in the figure that the two strong diffraction peaks 2θ of CF are 25.2°and 43.7°, and the characteristic peaks 2θ of MoS2 are 14.1°, 33.9°, 39.5°, and 59.1°. CF/MoS_2_ showed sharp diffraction peaks at 2θ of 33.9°, 39.5°, and 59.1°, which are characteristic peaks of MoS_2_. Therefore, it can be determined that MoS_2_ has been successfully grown onto the surface of the modified CF.

Figure 7 is the SEM image of the modified CF and MoS_2_/CF/PUE composites. It can be clearly seen from Figure 7a that MoS_2_ is generated in situ on CF, in Figure 7b, it can be observed that there are obvious dents and bumps on the PUE matrix. These bumps are well combined with the PUE matrix, which can reduce wear damage to a certain extent. At the same time, it can be observed that the CF particles are combined with the PUE matrix without shedding, because the contact area between the modified CF and the matrix increases, which increases the bonding between the two, thereby increasing the wear resistance of the composite material. It can be observed that there are many CF abrasive grains between the dents, which is because the CF molecular layers are covalently bonded to each other, so it is not easy for interlayer sliding to occur, but when the CF is ground up, it will form a graphite flake layer structure similar to MoS_2_, which plays a lubricating role between the friction interface and can effectively reduce the friction coefficient of PUE composites.

### 3.2. Performance of MoS_2_/CF/PUE Composites

The hardness of CF/PUE composite and MoS_2_/CF/PUE composites are shown in Figure 8, and it can be seen from the figure that the hardness of the modified PUE composites shows a trend of increasing and then decreasing with the increase in carbon content, and the maximum hardness value is 95.3 HA when the CF content is 0.5 wt%, which is 12% higher than that of pure PUE. Compared with Figure 8b, the hardness of the 0.5 wt% CF/PUE composite modified with MoS_2_ decreased by 4% compared with the unmodified 0.5 wt% CF/PUE composite, which is not a significant change. According to the histogram, the overall change in hardness of CF/PUE composites modified by MoS_2_ is less obvious than that of unmodified CF/PUE composites, which is due to the low hardness of MoS_2_ itself and the small amount of experimental dosage, resulting in an insignificant change in hardness.

In Figure 9a, the CF/PUE composites with 0.3 wt% CF content have good tensile properties. In Figure 9b, the tensile strength curve of the CF/PUE composites reached maximum at a CF content of 0.3 wt%, which is 43.81 MPa; the elongation at the break of the CF/PUE composites gradually decreased with the increase in CF content. As shown in Figure 10, the modified MoS_2_/CF/PUE composite has better tensile strength when the content of modified CF is 0.3 wt% and 0.5 wt%. The tensile strength of the composites increased first and then decreased with the increase in CF content, and the elongation at break also increased first and then decreased. From Figure 10b, the tensile strength is the best at the CF content of 0.3 wt% and 0.5 wt%, which are 38.44 MPa and 38.19 MPa, respectively, which are 53% and 52% higher than that of pure PUE elastomer, respectively. It can have good tensile properties in a wide range. From Figure 10b, when the CF content is 0.3 wt%, the elongation at break is the largest, which is 850%, which is 16.16% higher than that of pure PUE, and 25% higher than that of unmodified PUE.

This is mainly because MoS_2_ is a lamellar substance that easily slides between layers and cannot effectively load external tensile stresses, and the addition of MoS_2_ also causes the problem of discontinuity in the PUE matrix, thus slightly reducing the tensile strength of the composite after MoS_2_ modifies CF, but improving the toughness of the composite due to the lamellar structure of MoS_2_, as shown by the increase in elongation at the break.

Figure 11 shows the compression performance of the CF/PUE composite. It can be seen that CF/PUE composite with a CF content of 0.3 wt% and 0.5 wt% has better compression performance. Compared with the compression properties of the MoS_2_/CF/PUE composites in Figure 12, it can be seen from Figure 12b that the compressive strength of the modified MoS_2_/CF/PUE composites increases and then decreases with the increase in CF content, and the maximum compressive stresses are 2.32 MPa and 2.58 MPa for the CF content of 0.3 wt% and 0.5 wt%, respectively, which are 42% and 58% higher than those of the pure PUE elastomer. The compressive strength of CF PUE composites with 0.3 wt% and 0.5 wt% CF content decreased by 23% and 0%, respectively, compared to that of unmodified CF composites with 0.3 wt% and 0.5 wt% CF content.

The results showed that the compressive strength of the modified composites still maintained higher values at 0.3 wt% and 0.5 wt% of CF content, with a slight decrease at 0.3 wt% of CF content, which was also related to the lamellar structure of MoS_2_, the addition of MoS_2_ led to discontinuity in the PUE matrix, and was also affected by the inhomogeneity of the modified composition. The overall results are consistent with the tensile performance results.

The friction coefficient of MoS_2_/CF/PUE composites is shown in Figure 13. As can be seen from the graph, the modified composite friction coefficient shows a trend of decreasing and then increasing, and the minimum value is achieved when the CF content is 0.3 wt%, and the friction coefficient of the composite is 0.284 at this time, which is 59% lower than that of pure PUE, which is 0.688. In addition, MoS_2_/CF/PUE composites with a CF content of 0.5 wt% also have a low coefficient of friction.

Figure 14 shows the SEM image of the friction of the CF/PUE composite. From Figure 14a, we can see pure PUE elastomer abrasion marks. From Figure 14b, we can see, after adding a small amount of CF, a large number of grooves between the wear scars. In Figure 14c, fine wear debris and a small amount of detached CF can be observed. At this point, the surface is much flatter and the wear is much lower in comparison. This shows that the addition of CF effectively enhances the resistance of CF/PUE composite to steel ball cutting, makes the wear surface of the composite cleaner, and also greatly reduces the friction coefficient of the composite. In the initial period of frictional wear, the CF/PUE composite material is plastically deformed by frictional heat, and under the action of applying constant pressure, the PUE matrix produces stress concentration and microcracks, and large wear debris are produced under the cutting action of the steel ring, forming adhesive wear at the friction interface. After the friction and wear test was carried out for a period of time, CF particles entered the wear interface. On the one hand, they filled the surface of the sample, and on the other hand, they interacted with the wear debris to form an incomplete transfer film to protect the CF/PUE composite against the cutting of steel balls. However, at this time, the bonding force between the CF and the PUE matrix is weak, and the whole root will fall off under the shear action of the steel ball, forming a falling-off pit. When the friction wear test enters the stabilization phase, the CF interacts with the wear debris to form a complete transfer film at the friction interface. Large-scale peeling pits and aggregated CF can be observed in Figure 14d. This is due to the high overall content of CF, resulting in discontinuity of the PUE matrix and the agglomeration of CF, and falling-off during the friction and wear test. Because the CF surface is relatively smooth and there is some slurry, the CF and the PUE matrix are weakly combined, or defects are formed near the CF, and finally, the CF is separated from the PUE matrix to form a spalling pit.

Figure 15 shows the wear morphology of the modified CF/PUE composite material after 120 min of friction wear by dry friction with an applied load of 100 N, taken under high magnification. It can be seen that the overall wear debris of the modified composite material is finer and the wear condition is lighter. In Figure 15c,d, a large number of fine wear debris is distributed on the surface of the modified CF/PUE composite material, and the modified CF is uniformly dispersed in it, which effectively prevents the adhesive wear from affecting the material. At this time, the degree of friction is not deep, and the corresponding friction coefficient is not high, which is in good agreement with the experimental data.

Analysis shows that MoS_2_ is a solid lubricating material with a lamellar structure and has good self-lubricating properties. MoS_2_ easily reacts with air during the friction process, forming an oxide film at the friction interface to protect the PUE composite against the cutting action of steel balls, with the progress of the friction and wear test, a part of MoS_2_ was released to the friction interface to act as a lubricating medium, which transforms the friction motion between the steel ball and the PUE composite into the form of sliding between the MoS_2_ layers, effectively reducing the friction coefficient of the PUE composite. When its content is high, it destroys the continuity between the molecular chains inside the PUE and forms large-size wear debris, resulting in a higher coefficient of friction and a slightly lower coefficient of friction compared to pure PUE.

### 3.3. Wear Mechanism of MoS2/CF/PUE Composites

At the beginning of friction and wear (Figure 16), CF/PUE composites undergo plastic deformation due to friction heat. Under the action of constant pressure, the PUE matrix produces stress concentration and microcracks. Large pieces of wear debris are produced under the cutting action of the steel ball, forming adhesive wear at the friction interface. After a period of friction and wear test, CF particles enter the wear interface. On the one hand, the sample surface is filled, on the other hand, MoS_2_ is a solid lubricating material with a layered structure and excellent self-lubricating performance. MoS_2_ is prone to react with air during the friction process, forming an oxide film at the friction interface to protect the polyurethane composite material from the cutting action of the steel ball. As the friction and wear test progresses, a portion of MoS_2_ is released between the friction interface to serve as a lubricating medium, transforming the frictional motion between the steel ball and the polyurethane composite material into a sliding form between MoS_2_ layers, effectively reducing the friction coefficient of polyurethane composite materials.

## 4. Conclusions

MoS_2_ flakes were formed on the surface of CF by hydrothermal in situ generation modification. It was proved that MoS_2_ was successfully formed on the surface of CF by XRD and SEM. The CF reinforced PUE with a content of 0.3 wt% modified with MoS_2_ can obtain a composite material with better mechanical properties and a friction coefficient much lower than that of pure PUE. When the CF content is 0.3 wt%, the tensile strength increases by 53%, respectively, and the compressive strength increases by 42%, respectively, compared with pure PUE, and the mechanical properties are better in a wider range. In the form of sliding friction, the friction coefficient of the PUE/CF/MoS_2_ composite is reduced by 59% compared with pure PUE at 0.3 wt% of CF content. As a result, ternary composites with better friction properties can be produced by the in situ generation of MoS_2_ on CF.

## Figures and Tables

**Figure 1 materials-16-05773-f001:**
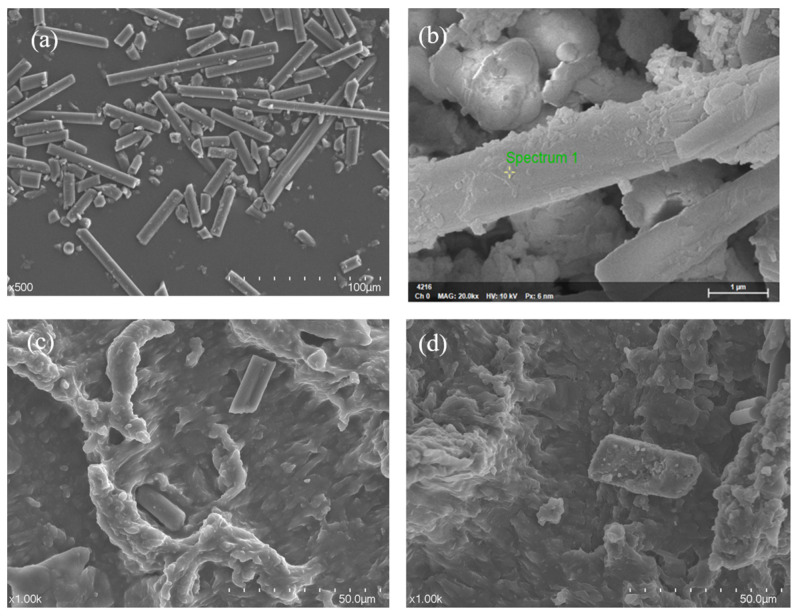
SEM diagram of CF and MoS_2_: (**a**) CF; (**b**) MoS_2_ on CF; (**c**) CF/PUE compsite without MoS_2_; (**d**) CF/PUE compsite with MoS_2_.

**Figure 2 materials-16-05773-f002:**
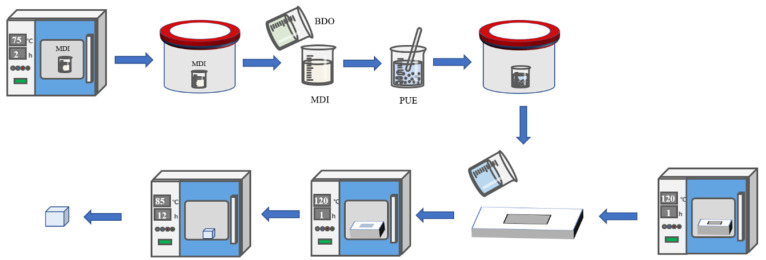
Process flow chart of PUE preparation.

**Figure 3 materials-16-05773-f003:**
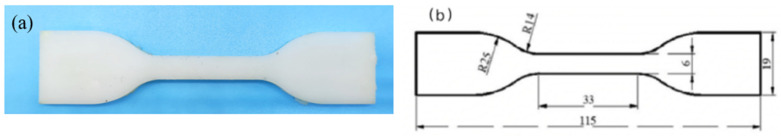
PUE composites tensile sample: (**a**) sample; (**b**) tensile specimen size.

**Figure 4 materials-16-05773-f004:**
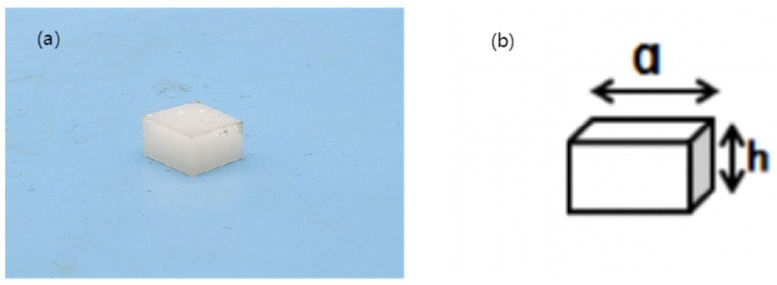
Compression sample of polyurethane composite: (**a**) sample; (**b**) compression sample size.

**Figure 5 materials-16-05773-f005:**
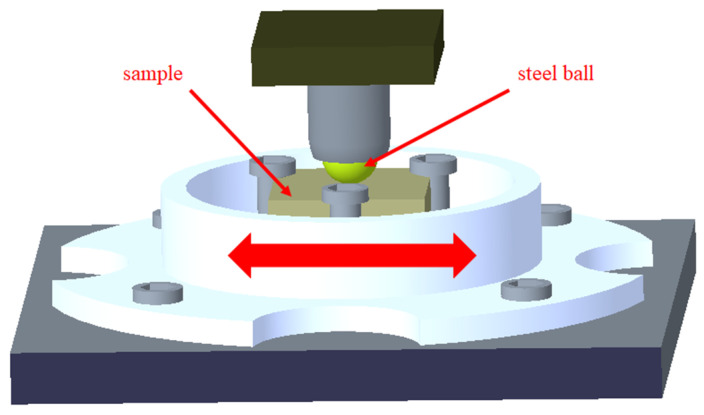
Matching of friction steel ball and sample.

**Figure 6 materials-16-05773-f006:**
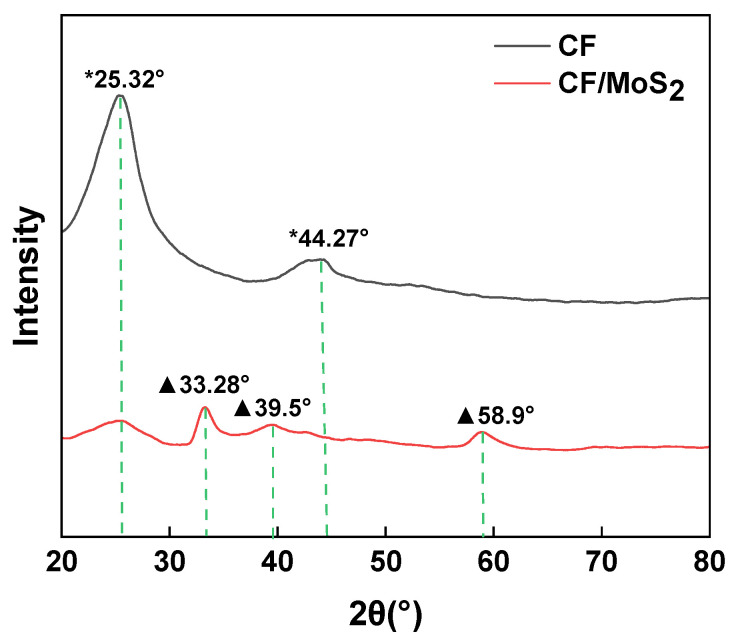
X-ray diffraction diagram of CF and molybdenum-modified CF.

**Figure 7 materials-16-05773-f007:**
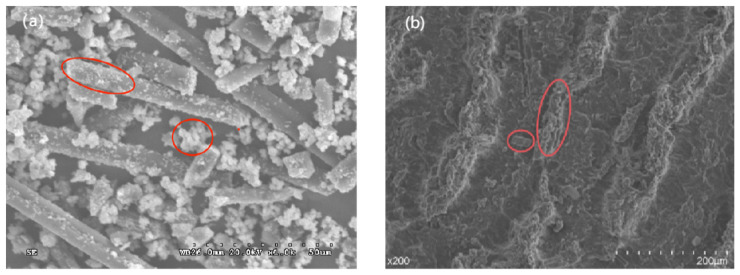
SEM diagram of modified CF and composite material: (**a**) modified CF; (**b**) MoS_2_/CF/PUE composite material.

**Figure 8 materials-16-05773-f008:**
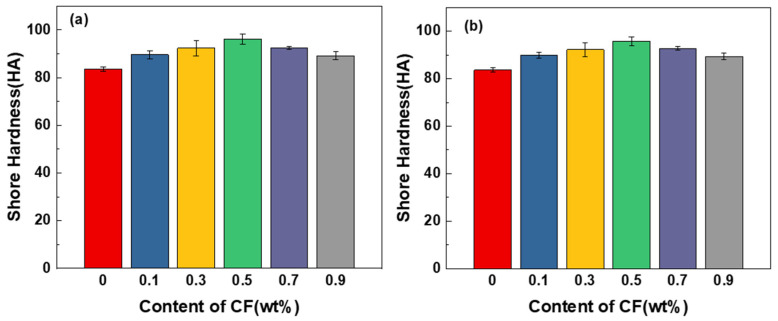
Shore hardness histogram: (**a**) CF/PUE composite; (**b**) MoS_2_/CF/PUE composite.

**Figure 9 materials-16-05773-f009:**
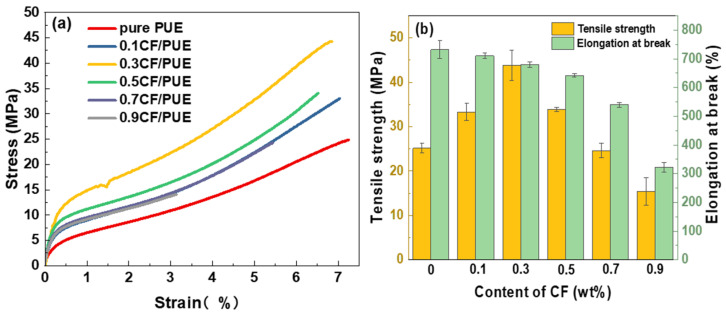
Tensile properties of CF/PUE composites: (**a**) tensile stress-strain curve; (**b**) tensile strength histogram and elongation at break histogram.

**Figure 10 materials-16-05773-f010:**
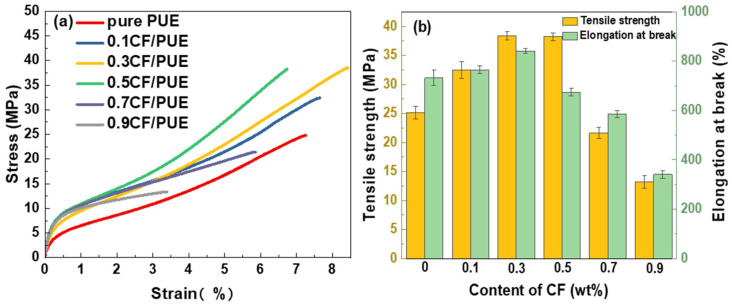
Tensile properties of MoS_2_/CF/PUE composite: (**a**) tensile stress and strain curve; (**b**) tensile strength histogram elongation at break histogram.

**Figure 11 materials-16-05773-f011:**
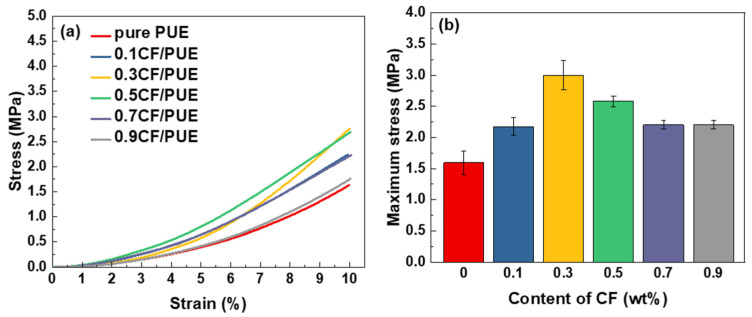
Compression properties of CF/PUE composites: (**a**) compressive stress-strain curve; (**b**) maximum compressive stress histogram.

**Figure 12 materials-16-05773-f012:**
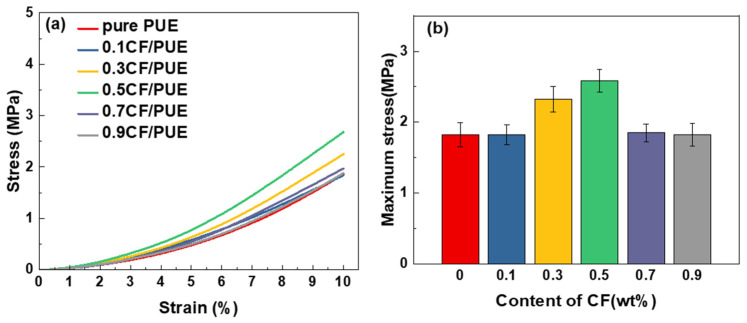
Compression properties of MoS_2_/CF/PUE composite: (**a**) compressive stress and strain curve; (**b**) maximum compression stress histogram.

**Figure 13 materials-16-05773-f013:**
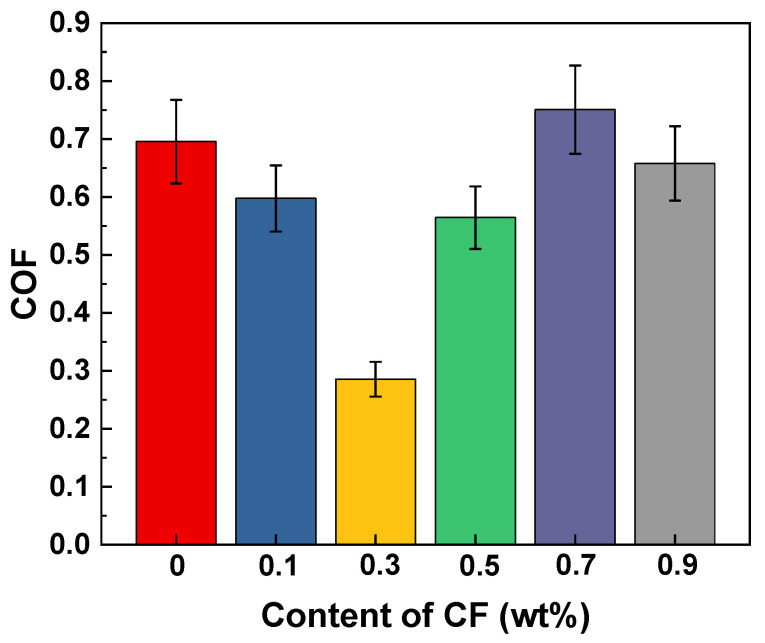
Friction coefficient of PUE/CF/MoS_2_ composites.

**Figure 14 materials-16-05773-f014:**
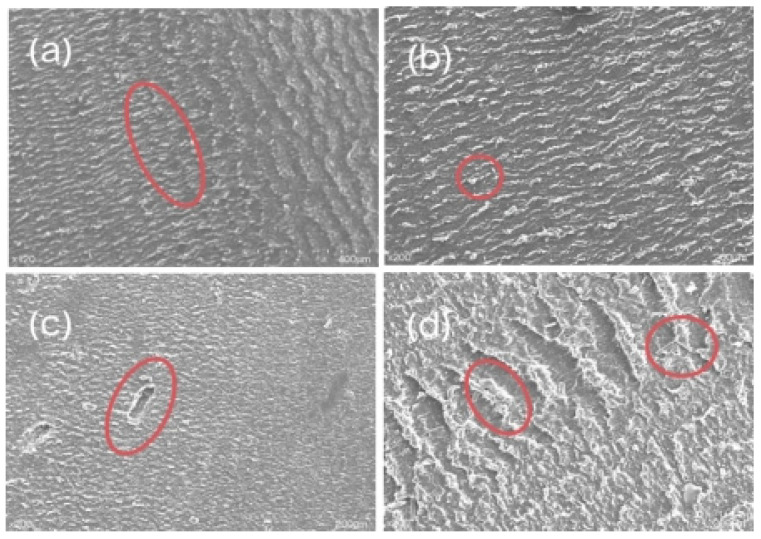
Friction SEM images of CF/PUE composites: (**a**) pure PUE friction sample; (**b**) 0.1CF/PUE friction sample; (**c**) 0.5CF/PUE friction sample; (**d**) 0.9CF/PUE friction sample.

**Figure 15 materials-16-05773-f015:**
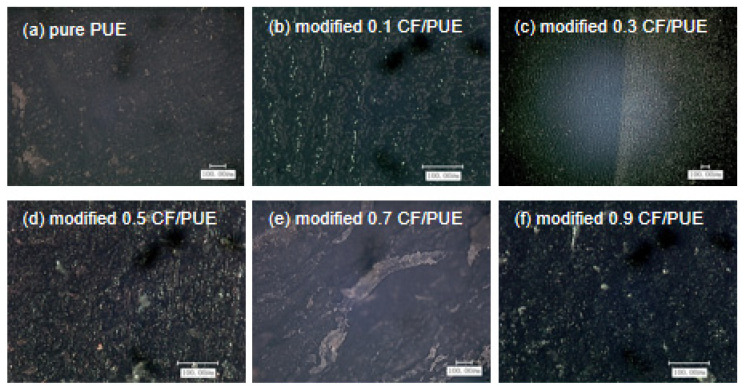
Wear morphology of a modified CF/PUE composite.

**Figure 16 materials-16-05773-f016:**
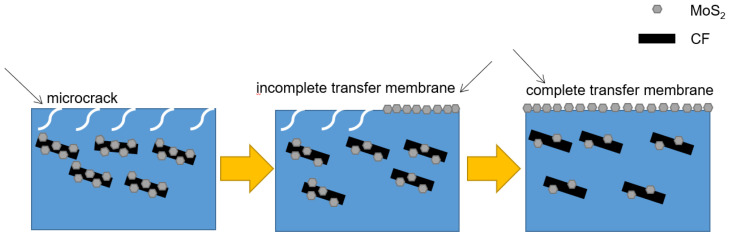
Wear mechanism of PUE/CF/MoS2 composites.

**Table 1 materials-16-05773-t001:** Equipment materials used in the experiment.

Experimental Materials and Equipment	Producer
MDI prepolymer	Shanghai Hecheng Polymer Technology Co., Ltd. CHN city, country
1,4 butanediol (BDO)	Guangdong Yuemei Chemical Co., Ltd. CHN
Carbon fiber	Hangzhou Gaoke composite material Co., Ltd. CHN
Thiourea (CH4N2S)	Aladdin reagent (Shanghai) Co., Ltd. CHN
Molybdic acid (H8MoN2O4)	Aladdin reagent (Shanghai) Co., Ltd. CHN
Nitric acid	Aladdin reagent (Shanghai) Co., Ltd. CHN
Absolute ethanol	Xilong Chemical Co., Ltd. ShanTou, CHN
Electronic balance: ZA2054AS	Shanghai Zanwei weighing instrument Co., Ltd. CHN
Ultrasonic cleaning machine: KQ2200E	Kunshan Ultrasonic Instrument Co., Ltd. CHN
Blast drying oven: XGQ-200	Yuyao Xingchen Instrument Factory, CHN
Vacuum pump: XP135	Dingsheng Vacuum Equipment Co., Ltd. CHN
Vacuum tank: XP135	Dingsheng Vacuum Equipment Co., Ltd. CHN
Digital display electric mixer: JJ-1A 100W	Fangke instrument (Changzhou) Co., Ltd. CHN
Friction and wear tester: UMT-3	Beijing Yicheng Hengda Technology Co., Ltd. CHN
Universal tensile tester: PT-307,	Dongguan pusaite testing equipment Co., Ltd. CHN
High-speed camera: VW-9000,	Keans (China) Co., Ltd.
Shore hardness tester: type A,	Yangzhou process test Machinery Co., Ltd. CHN
D8ADVANCE X-ray diffractometer	Bruker AG, Germany
G-301 scanning electron microscope	Hitachi, Ltd. JP

## Data Availability

Not applicable.

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
