# Peer review of "In Situ Formation of MoS2 on the Surface of CF to Improve the Tribological Properties of PUE"

_materials, 2023, doi:10.3390/ma16175773_

Round 1

Reviewer 1 Report

The work presented in this manuscript displayed interest. However, there are some issues have to be addressed as listed.

1. The abstract need to be revised focusing what is new and highlighting new with numbers.

2. Introduction is poor need citation of potential reported published in last three years 2021-2023 and i recommend citation of this report

Materials Science and Engineering: B 274 (2021) 115460

3. What is the dimension of the carbon fiber used and SEM image is required clarifying the surface and dimension.

4. What is the dimension of MoS2 and shape?

5. What was the content of MoS2 on CF/MoS2 composite ?

6. What is the dispersion of CF/MoS2 composite in PU; it is need clear investigation .

7. The quality of figures need to be improved.

8. Comparative study is required.

English need revision. 

Reviewer 2 Report

The authors presented an article titled “In situ formation of MoS2 on the surface of CF to improve the tribological properties of PUE”. This article is covered by the "Materials" journal. However, the article will be ready for publication after a major revision. Comments are listed below.

1.      Author names and article title should be checked. (Different in the article, different in the referee interface)

2.      A sentence about numerical results can be written in the summary section.

3.      The disadvantages of MoS2/CF composites should be mentioned in the introduction.

4.      What is the novelty of this work? The difference from similar studies in the literature should be explained.

5.      Table and figure numbers are incorrect throughout the article. For example Figure 1, Figure 2.... and Table 1, Table 2....)

6.      The table on page 2, line 69 has no title. It should be added. In addition, the table should be referenced in the text.

7.      Figure 1-1 should be cited in the text.

8.      During preparation of MoS2/CF (0.1, 0.3, 0.5, 0.7, 0.9%) of CF by weight or volume? How were these rates determined?

9.      How were the hardness and tensile specimens prepared? Details should be added to the Materials and Methods section.

10.  According to which standards were the parameters used in the Tribological Performance Test determined?

11.  The resolution of the images given in Figure 2-2 is very low and the texts are not readable. The resolution should be increased and the scale should be highlighted.

12.  Results sections are generally devoid of discussion. It should be discussed and compared with similar studies in the literature.

13.  The resolution of the images in Figure 2-9 is too low. It should be increased. In addition, a scale should be given for each image.

14.  The resolution of the images in Figure 2-10 is too low. It should be increased.

15.  Apparently, conclusions are just observations. The explanations given for the conclusions of the article need to be checked thoroughly.

16.  The article contains numerous typographic and language errors. It should be corrected.

17.  The article should be rearranged by taking into account the journal writing rules and citation rules.

Reviewer 3 Report

Clear thematic work with a clear description of the problem, experiments and results. The data obtained are useful and confirm the previously discovered patterns on the effect of small additives of dispersed fillers, incl. CF on the properties of composites.

However, there are a few things to note:

1)    Why was a high-speed camera used to study wear morphology?

2)    What accelerating voltage and type of detector were used for the SEM?

3)    For morphology comparison, an image of a PU-CF composite without MoS2 should be added. Moreover, the presented magnification is not enough for a qualitative assessment of the interfacial interaction of PU with CF.

4)    The experimental data should be rounded correct. Indication of hundredths of a percent in the values of elongation and strength with such measurement errors is not correct.

Minor flaws:

5)    During all text: There must be a space between the number and the dimension (4 h, 10 °Ð¡ etc.)

6)    Line 90. Liter must be written as capital L, not a small letter. (mL instead ml).

7)    The colors of the lines in figures 2-3 - 2-8 should be the same for all samples so that it is easy to understand what corresponds to what.

8)    On fig. 2-9, the size scale is very poorly visible. This needs to be fixed.

Reviewer 4 Report

The manuscript "In situ formation of MoS2 on the surface of CF to improve the tribological properties of PUE" has an interesting subject of the research field.

The presented results are interesting and can be useful, and for this reason the work can be considered for publish.

However some corrections and observations must be considered:

- the introduction must be more applied and need add some valuable lost references.

- the references must write with MDPI standard respect.

- the experimental part need attention because some reagents and equipment are used in work, but not mentioned in table.

- some figures need author attention: fig. 2-2. SEM images must be at same magnitude (and contrast if it is possible); fig. 2-9. images need a dimension  scale; fig. 2-10. need  be more readable;

- the conclusion section must be remake.

Round 2

Reviewer 2 Report

The authors have completed the necessary revisions. I have no further questions. The article can be evaluated for publication in its final form.